# Prioritising older individuals for COVID-19 booster vaccination leads to optimal public health outcomes in a range of socio-economic settings

Ioana Bouros[1], Edward M. Hill[2,3,4], Matt J. Keeling[2,3,4], Sam Moore[5], Robin N. Thompson[6] *

1 Department of Computer Science, University of Oxford, Oxford, United Kingdom, 2 Mathematics Institute, University of Warwick, Coventry, United Kingdom, 3 Zeeman Institute for Systems Biology and Infectious Disease Epidemiology Research (SBIDER), University of Warwick, Coventry, United Kingdom, 4 School of Life Sciences, University of Warwick, Coventry, United Kingdom, 5 Lancaster Medical School, Lancaster University, Lancaster, United Kingdom, 6 Mathematical Institute, University of Oxford, Oxford, United Kingdom

☯ These authors contributed equally to this work.
* robin.thompson@maths.ox.ac.uk

**Data Availability Statement:** The computing code used to perform the analyses in this article is available in the following GitHub repository: https://

## Abstract

The rapid development of vaccines against SARS-CoV-2 altered the course of the COVID-19 pandemic. In most countries, vaccinations were initially targeted at high-risk populations, including older individuals and healthcare workers. Now, despite substantial infection- and vaccine-induced immunity in host populations worldwide, waning immunity and the emergence of novel variants continue to cause significant waves of infection and disease. Policy makers must determine how to deploy booster vaccinations, particularly when constraints in vaccine supply, delivery and cost mean that booster vaccines cannot be administered to everyone. A key question is therefore whether older individuals should again be prioritised for vaccination, or whether alternative strategies (e.g. offering booster vaccines to the individuals who have most contacts with others and therefore drive infection) can instead offer indirect protection to older individuals. Here, we use mathematical modelling to address this question, considering SARS-CoV-2 transmission in a range of countries with different socio-economic backgrounds. We show that the population structures of different countries can have a pronounced effect on the impact of booster vaccination, even when identical booster vaccination targeting strategies are adopted. However, under the assumed transmission model, prioritising older individuals for booster vaccination consistently leads to the most favourable public health outcomes in every setting considered. This remains true for a range of assumptions about booster vaccine supply and timing, and for different assumed policy objectives of booster vaccination.

github.com/I-Bouros/warwick-covid-transmission.
All code was written in Python (compatible with
versions 3.7-3.9).

**Funding:** This research was funded by the World
Health Organization through a grant from the SAGE
Working Group on COVID-19 Vaccines (IB, EMH,
MJK, SM, RNT). This grant included funding for
IB's salary for the duration of the project, and
partially funded RNT's and SM's salary. EMH, MJK,
SM and RNT acknowledge the support of the
JUNIPER partnership, funded by MRC (grant
number MR/X018598/1), to which they are linked.
The funders had no role in study design, data
collection and analysis, decision to publish, or
preparation of the manuscript.

**Competing interests:** The authors have declared
that no competing interests exist.

## Author summary

Computational modelling is used increasingly to assess interventions against infectious
diseases. For COVID-19, booster vaccines are being deployed worldwide to mitigate the
effect of waning immunity following past infections and vaccinations, and to protect
against new SARS-CoV-2 variants. However, booster vaccine availability is limited, mak-
ing it necessary to consider who should receive booster vaccinations. It is essential that
booster vaccine prioritisation is considered not only in the context of high income coun-
tries, but also low and middle income countries where the population age structure is very
different. We have therefore developed a computational model to guide booster vaccina-
tion prioritisation in different socio-economic settings. We show that vaccinated popula-
tions can have different compositions between countries, even when identical booster
vaccination strategies are adopted. Despite these differences, our modelling results suggest
that booster vaccination strategies prioritising the oldest individuals lead to the best possi-
ble public health outcomes, due to the substantial skew in the risk of severe outcomes of
infection towards older (and other vulnerable) hosts. This remains true even when quanti-
fying the extent of premature, rather than absolute, mortality. This research therefore sup-
ports policy advice recommending prioritisation of older individuals for booster vaccines
in countries worldwide when vaccine supply is limited.

## Introduction

In the early stages of the COVID-19 pandemic, non-pharmaceutical interventions (NPIs) were
introduced worldwide to reduce transmission and to limit the public health impacts of SARS-
CoV-2 [1–3]. While NPIs played a key role in reducing the burden on healthcare systems, the
development of vaccines changed the longer term course of the COVID-19 pandemic. Vacci-
nation reduced the number of individuals experiencing the most severe consequences of infec-
tion [4] and allowed NPIs to be relaxed. Vaccines produced by various manufacturers,
including Oxford-AstraZeneca, Pfizer-BioNTech and Moderna, have been administered
around the world.

Unfortunately, the protection afforded by COVID-19 vaccines is not permanent, with
immunity waning in the months following vaccination [5–9]. In addition, the emergence of
novel SARS-CoV-2 variants affects the effective level of immunity in host populations [10–13].
To counter waning immunity and the emergence of novel variants, re-formulated booster vac-
cines designed to protect against the latest variants have been deployed. While booster vaccines
are a useful resource, the capacity to deliver booster vaccines is inherently limited, so it is
unlikely that entire populations can receive booster vaccinations frequently, particularly over
long time periods. Regular booster vaccination of the entire population is also unlikely to be
cost-effective, such that health resources are better utilised elsewhere. A key question for public
health policy makers going forwards is therefore who should be prioritised for booster
vaccination.

Mathematical modelling has been used to explore the effects of a range of interventions
against SARS-CoV-2 [14–21], with modelling projections and estimates of epidemiological
metrics such as the time-dependent reproduction number (the "R number") [22–25] often at
the centre of the news agenda in the acute phase of the COVID-19 pandemic. Models have
been used to investigate optimal vaccination targeting. Some previous studies have suggested
that, in certain scenarios and with some control objectives, vaccination strategies that aim to
prevent transmission and reduce infection prevalence can be more effective than strategies

that prioritise protecting the most vulnerable individuals directly [26–29]. In contrast, other research has indicated that prioritising vaccination of vulnerable individuals, who are most likely to experience severe consequences of infection if they become infected, can be the optimal approach [28,30–32].

As with other interventions, the effectiveness of vaccination strategies targeting specific subgroups of a population depends on associated heterogeneities (for example, the relative transmission rates and risks of severe infection outcomes associated with vulnerable and non-vulnerable individuals) and, crucially, on the objectives of policy makers [33]. Early in the COVID-19 pandemic, Moore *et al.* [31] studied age-based vaccination strategies in the UK. They showed that, if the objective was to minimise COVID-19 deaths, prioritising older age groups for vaccination was likely to be the optimal strategy in that setting. Similarly, Keeling *et al.* [30] considered the impact of COVID-19 vaccination on numbers of deaths in England and found that vaccinating the vulnerable is a particularly effective strategy when there is a high prevalence of infection in the host population.

While previous studies have generated useful insights into the effectiveness of different COVID-19 vaccination strategies, particularly early in the COVID-19 pandemic, there is a need to consider the impacts of different booster vaccination strategies going forwards now that there is substantial infection- and vaccine-induced immunity in countries worldwide. In addition, it is necessary to investigate whether the optimal booster vaccination strategy in countries such as the UK is also optimal in countries with different population structures, with a particular evidence gap concerning optimal vaccine prioritisation in low and middle income countries [34]. For example, lower income countries tend to have a higher proportion of younger individuals than higher income countries [35]. Since older individuals are generally more vulnerable to the most severe effects of SARS-CoV-2 infection than younger individuals [36,37], population age structure has implications for population-level transmission and health risks, and potentially for optimising interventions. Similarly, differing social structures, such as the existence of multi-generational households in many lower income countries, mean that older populations in different countries may experience different levels of exposure to infection.

In this study, we extend the model of SARS-CoV-2 transmission and vaccination presented by Moore *et al.* [38] and consider a wave of infections caused by a novel SARS-CoV-2 variant. We assume that there is substantial immunity in the population due to previous infection and vaccination. We consider a setting in which a limited number of booster vaccines are available and investigate the effects of six different age-based booster vaccination prioritisation strategies in eight countries chosen to reflect a range of socio-economic backgrounds and population structures. We not only explore the effects of the different booster vaccination strategies on projected numbers of deaths, but we also consider Years of Life Lost (YLL)–a measure of premature mortality that accounts not only for the number of deaths but also the ages of individuals when they die. In each scenario that we consider using our transmission model, we find that the substantial skew in the risk of severe outcomes of SARS-CoV-2 infection towards older individuals leads to an optimal strategy of prioritising the oldest individuals for booster vaccination. This strategy was used in many countries during the acute phase of the COVID-19 pandemic, and our modelling results support its continued use for prioritising booster vaccinations going forwards.

## Methods

### Epidemiological model

In our analyses, we extend a previously published age structured compartmental model of SARS-CoV-2 transmission and vaccination [38] (see S1 Text). The model is deterministic,

comprising of a system of ordinary differential equations. Individuals are divided into compartments based on the following characteristics, which we explain in more detail below: i) age; ii) current infection status, and; iii) immune status (prior to any current infection).

There are 16 age groups in the model. The first 15 age groups partition individuals aged between 0–74 years old into five-year intervals (0–4 years old, 5–9 years old, . . ., 70–74 years old). The final age group represents individuals who are 75 years old and over.

Individuals are classified according to their current infection status: either Susceptible, Exposed, Infectious (either symptomatic or asymptomatic) or Recovered. We divide the exposed class into multiple compartments in series. This is so that, in the stochastic model that is analogous to the deterministic model that we use, individuals would spend a gamma-distributed time period in the exposed class. Gamma distributions have previously been found to provide more realistic representations of epidemiological periods than exponential distributions, despite the fact that epidemiological periods are often assumed to be drawn from exponential distributions [39–41].

Finally, individuals' immune statuses (for exposed or infectious individuals, an individual's immune status refers to their immune status prior to their current infection) are tracked. We divide individuals into groups corresponding to different levels of immunity, reflecting their infection and vaccination history (specifically, the time period since they were last infected or vaccinated). Individuals are therefore assigned to one of the following groups: Vaccinated (but have not received a booster vaccine), Boosted, Partially waned, Fully waned, or Unvaccinated (and not previously infected). Individuals in the partially waned group correspond to those who have received a booster vaccination and/or have previously been infected, but sufficiently long ago that their immunity has begun to wane. Those in the Fully waned group are either vaccinated (but not boosted) individuals whose immunity has begun to wane, or individuals who have received a booster vaccination or been infected previously but with a relatively long time period (on average) having elapsed since the booster vaccination was administered (or since they were infected) resulting in more substantial waning of immunity. The Unvaccinated class includes individuals with no vaccination or infection history, as well as those who previously received a vaccination or were infected but long enough ago that little immunity remains. The extent to which individuals with different immune statuses are assumed to be protected from infection, developing symptoms, becoming hospitalised or dying is explained later (see Table 1).

A schematic illustrating how individuals transition between compartments corresponding to different infection and immunity statuses is shown in Fig 1A. Each model projection is run for a duration of 150 days, and each individual is assumed to remain in the same age group throughout each projection.

## Public health outcomes derived from model projections

While only infections are tracked explicitly in the transmission model (rather than hospitalisations or deaths), the number of deaths throughout each model projection is estimated based

**Table 1. The assumed effects of vaccination on the risk and outcomes of infection.** The overall reduction in the risk of infection, symptomatic infection, hospitalisation and death assumed in our transmission model, for a susceptible individual of each immune status (represented by each row) relative to an unvaccinated individual with no existing immunity. These values are based on those assumed in the UK Health Security Agency vaccination surveillance reports [49].

|  | Infection | Symptoms | Hospitalisation | Death |
|---|---|---|---|---|
| **Vaccinated** | 30% | 40% | 80% | 80% |
| **Boosted** | 40% | 60% | 90% | 95% |
| **Partially Waned** | 20% | 40% | 85% | 85% |
| **Fully Waned** | 5% | 20% | 60% | 60% |

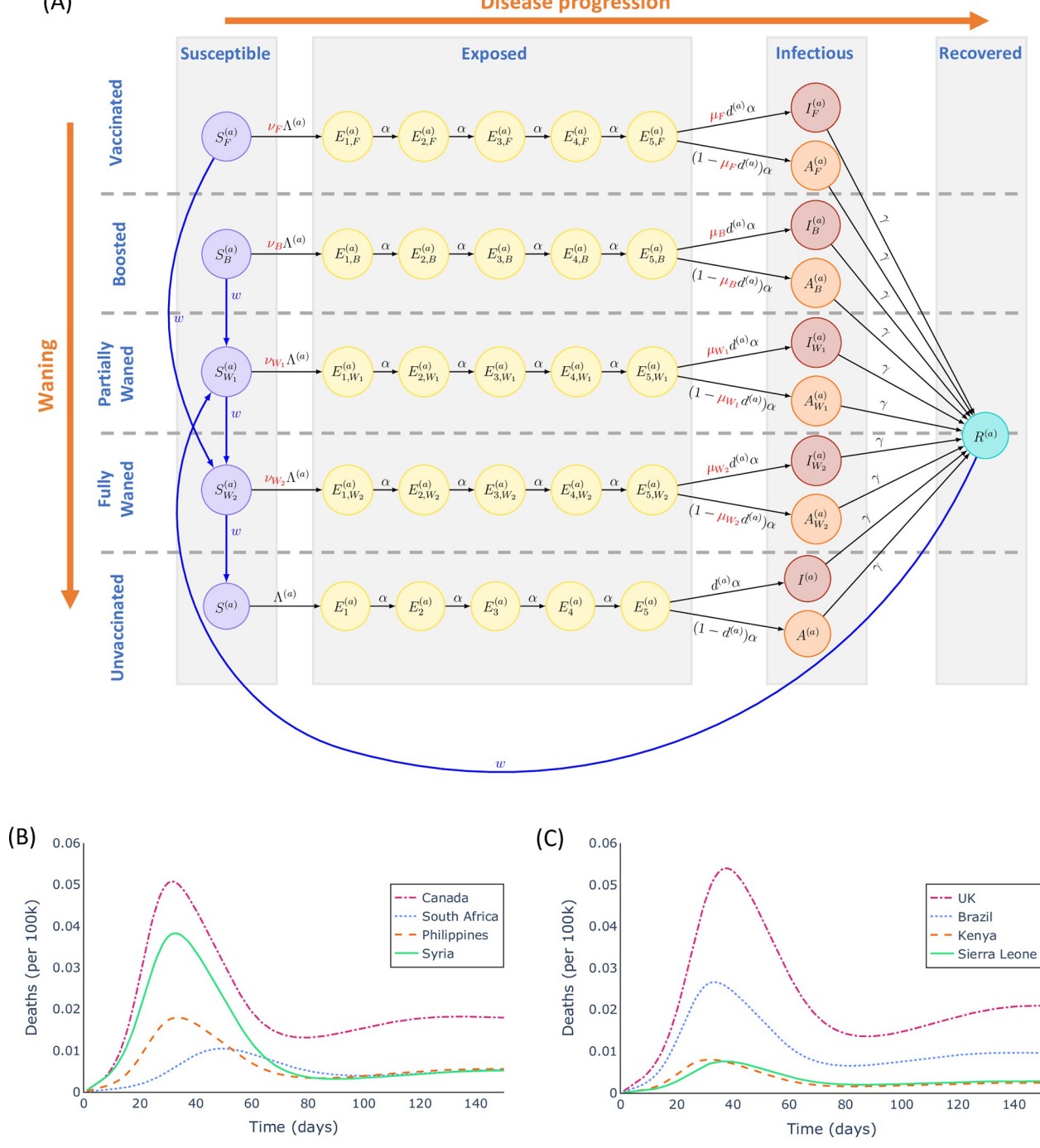

**Fig 1. The mathematical model of SARS-CoV-2 transmission and vaccination used in our analyses.** A. Schematic illustrating the compartmental structure of the model. Individuals are divided according to both their current infection status (left to right) and their immune status prior to any current infection (top to bottom). In addition, individuals are divided according to their age group (as described in the text, denoted by the superscript *a*). The variables shown in red relate to booster vaccine effectiveness (see S1 Text). B-C. Outputs from our transmission model with the baseline parameter values in the absence of any booster vaccination strategy. The numbers of deaths each day (per 100,000 individuals in the population) are shown for a range of countries spanning four different socio-economic statuses.

on the number of symptomatic infections in each age group. Specifically, the projected number of deaths on any given day is calculated based on: i) the proportion of symptomatic infected individuals who experience severe disease and are hospitalised; ii) the proportion of hospitalised individuals who die; iii) a probability distribution characterising the time period between an individual developing symptoms and being admitted to hospital, and; iv) a probability distribution characterising the time period between an individual being admitted to hospital and dying. Additional information about this calculation is provided in S1 Text.

For calculating Years of Life Lost (YLL) in each country due to COVID-19, we weight each death by the expected time period until the individual would have died in the absence of SARS-CoV-2 infection. In other words, for an individual who dies at age $x$, the contribution to the YLL is the expected remaining duration of the individual's life given that they have already survived to age $x$, as calculated from WHO country-specific life tables [42].

## Model parameterisation

We generate model projections for eight countries chosen to reflect a range of different socio-economic backgrounds. Those countries are: the UK, Canada (high income); Brazil, South Africa (higher-middle income); Kenya, the Philippines (lower-middle income), and; Sierra Leone, Syria (low income). In our model, the values of some parameters are assumed to be identical in all countries, and the values of other parameters are assumed to be specific to individual countries.

The core infection parameter values are assumed to be the same in all countries. These include age-dependent parameters characterising susceptibility to infection ($\beta^{(a)}$) and the probability that an infected individual develops symptoms ($d^{(a)}$), as well as the relative infection rate for asymptomatic infectious individuals compared to symptomatic infectious individuals ($\tau$) [43]. Estimates of the overall infectiousness parameter ($\omega$), the infectious period ($1/\gamma$) and the probability of a symptomatic infected individual developing severe disease ($P_{IH}^{(a)}$) are obtained by fitting the model to data describing age-stratified numbers of cases, hospitalisations and deaths in the UK between June and September 2022 using Markov chain Monte Carlo (MCMC; for further details, see S1 Text). This time period corresponds to the third wave of infection after the emergence of the Omicron variant in that country. To account for uncertainty in the values of these model parameters, in our analyses we consider a range of estimated parameter sets by sampling from the joint posterior distribution (corresponding to different steps of the MCMC chain; the stationary distribution of this chain corresponds to the target posterior distribution that we aim to sample from) and use the entire range of parameter values in our analyses.

We set the rate at which immunity wanes so that the expected time period that an individual spends at each immunity level is $1/w = 90$ days. The expected time period from an individual receiving a booster vaccination to becoming fully susceptible is therefore approximately nine months in the model. Although there is substantial uncertainty in this value, and some residual protection may last for long periods in practice, previous studies indicate that the duration of protection against SARS-CoV-2 infection following infection or vaccination is likely to be of this order of magnitude [44–48].

Infection- or vaccine-induced immunity has the potential to affect multiple aspects of an individual's infection. For example, vaccination or previous infection may reduce the risk of an individual becoming infected in the first place (host susceptibility), and it may also reduce the risk of an individual experiencing severe outcomes if they become infected. In our transmission model, we account for multiple effects of vaccination by altering the relevant rates in the model according to individuals' immune statuses prior to any current infection. We

present the assumed baseline levels of protection against infection, symptoms, hospitalisation and death for individuals with different immune statuses in Table 1. We note that these protection levels represent overall effects. For example, for a vaccinated individual to die due to SARS-CoV-2 infection, they need to first become infected, then develop symptoms, and then be hospitalised, all of which are less likely as a result of vaccination. The value of 80% in the top right of Table 1 reflects the overall reduction in the risk of a vaccinated individual dying due to SARS-CoV-2 infection compared to an unvaccinated individual, accounting for all of the effects of vaccination. For further details about how these protection levels are incorporated into our model parameterisation, see S1 Text.

In each country, the age structure of the population is set according to demographic estimates from the World Bank [35]. In addition, we set contact rates between individuals of different ages using the country-specific contract matrices calculated or inferred by Prem *et al.* [50]. The initial conditions in each model projection are determined based on estimates of the numbers of previous infections and vaccinations in each country. Numbers of existing partial, full initial dose, and recent booster vaccinations are sourced from Our World in Data [51]. Numbers of previous infections, as well as country-specific parameter values characterising the overall proportion of hospitalised individuals who die, are estimated for each country by fitting our model to excess mortality data from the Institute for Health Metrics and Evaluation (IHME) [52] up to the end of 2022. Further details are provided in S1 Text.

Together with the uncertainties in our core infection parameters, it is challenging to obtain accurate COVID-19 mortality estimates for many countries. To account for this additional level of uncertainty, for each infection parameter set used, we draw multiple samples uniformly at random between bounds from the excess mortality data provided by the IHME [52] and conduct our model fitting separately for each sample. This generates a range of different plausible initial conditions (along with the corresponding core infection parameter values). In our model projections, using this approach we generate projections separately for 100 different parameter sets, thereby obtaining a range of possible projections that we represent using 95% prediction intervals (i.e., intervals in which 95% of all simulated outcomes lie).

### Booster vaccination strategies

We use the model to explore the impacts of a range of different age-based booster vaccination prioritisation strategies on the projected numbers of deaths and YLL in each of the eight countries selected. The strategies explored were:

- Strategy 1: Decreasing age order (first individuals aged 75+, then 70–74, then 65–69, and so on).

- Strategy 2: First individuals aged 60–74, then 75+, and then in decreasing age order.

- Strategy 3: First individuals aged 60–74, then 50–59, then 75+, and then in decreasing age order.

- Strategy 4: First individuals aged 50–59, then 60–74, then 75+, and then in decreasing age order.

- Strategy 5: First individuals aged 20–49, then 50–74, then 75+, and then in decreasing age order.

- Strategy 6: First individuals aged 20–49, then 75+, then 50–74, and then in decreasing age order.

The premise behind the first strategy is that administering booster vaccines to the oldest individuals first protects many of the most vulnerable people in the population directly. The

final strategy tests the alternative hypothesis that vaccinating individuals who have large numbers of contacts may provide indirect protection for the oldest individuals. Strategies 2–5 test different variations on these two approaches, with the first four strategies all involving older individuals generally being prioritised for booster vaccination.

When, for example, booster vaccines are administered to individuals aged 50–59, individuals are chosen for vaccination from the corresponding age groups in the model (in that case, an equal proportion of individuals aged 50–54 and 55–59 receive booster vaccinations). We note that, in addition to the booster vaccinations administered in our model simulations, when the initial conditions of the model are determined for each country, some individuals have received booster vaccines soon before the start of our model projections; these individuals are also in the $S_B^{(a)}$ class of the model at the beginning of our projections.

In each model projection, to reflect limited vaccine availability and deployment capacity, we assume that the number of available booster vaccines is equal to 10% of the country's entire population size in our baseline analyses (we consider other vaccine availability values in supplementary analyses). Within each age group, we assume a maximum vaccine uptake level of 90% (that is, at most 90% of individuals in any age group can become vaccinated; alternative assumptions are also considered in supplementary analyses). Booster vaccines are assumed to be administered to individuals irrespective of their previous immune status, and are assumed to be administered at the start of each model projection. In the model, booster vaccination generally involves moving susceptible individuals from the other classes into the booster vaccinated ($S_B^{(a)}$) class. However, if a recovered individual receives a booster vaccination, they remain in the $R^{(a)}$ class, since recent infection is assumed to afford higher protection than booster vaccination.

## Results

### Booster vaccination in different countries

The age structure of the host population varies substantially between different countries. In general, high income countries tend to have a larger proportion of older individuals than low income countries. We therefore began by assessing the individuals to whom booster vaccines are administered under each of the booster vaccination strategies considered. These results are shown for the UK in Fig 2.

The focus of Strategy 1 is protecting the oldest individuals in the population (Fig 2A). Strategies 2–4 also aim to vaccinate many of the older individuals in the population. However, in the UK, because of the large number of individuals aged 50–59 and 60–74, in our baseline analyses (with a limited vaccine supply) Strategies 2–4 fail to protect the oldest individuals in the population (those aged 75+; Fig 2B–2D). Strategies 5–6 involve distributing booster vaccines among individuals aged 20–49, who typically have large numbers of contacts compared to individuals in older age groups (Fig 2E and 2F).

However, the ages of the vaccinated population under each strategy vary between countries. In Sierra Leone, for example, all of Strategies 1–4 involve vaccinating individuals in the 75 + age group (Fig 3). This is because the relatively small proportion of older individuals in that country permits vaccines to be distributed to the oldest members of the population under each of these strategies. In fact, in Sierra Leone, Strategies 1–4 have identical effects.

### Public health outcomes

To consider a wave of infections due to a novel SARS-CoV-2 variant, we generated model projections under each vaccination strategy for a time period of 150 days. We recorded the total number of projected deaths in each scenario. As described in the Methods, model projections

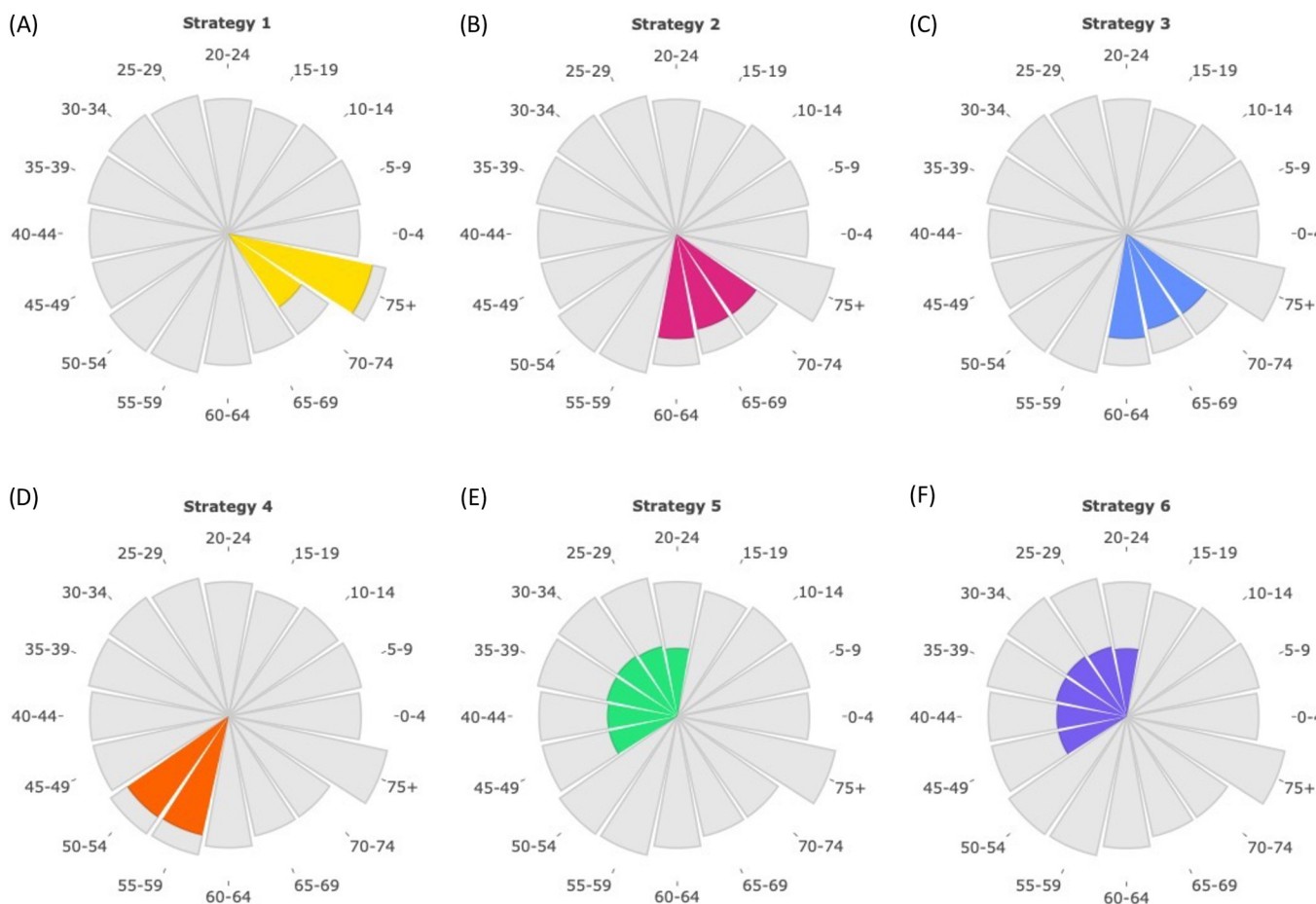

**Fig 2. The ages of individuals in the UK who receive booster vaccinations under each booster vaccination strategy.** The area of each wedge is proportional to the number of individuals in the UK in that age group. The coloured regions reflect the numbers of individuals in each age group who receive booster vaccinations under each booster vaccination strategy. The first four strategies generally involve prioritising older individuals for booster vaccination, whereas the final two strategies instead target younger individuals for booster vaccination. Booster vaccinations are administered at the start of our model projections.

were generated for a range of estimated parameter sets (100 in total), leading to a range of projected numbers of deaths under each booster vaccination strategy. Considering the mean number of deaths across these projections in each country under each booster vaccination strategy indicated that, in every country that we considered, Strategy 1 was projected to lead to the fewest deaths (Fig 4A). Due to the high proportion of older individuals in high income countries compared to low income countries, a particularly large number of deaths (as a proportion of the total population size) was projected to occur in high income countries. This is both because older individuals are more likely to die if infected by SARS-CoV-2, but also because the limited number of available booster vaccines does not allow as high a proportion of individuals aged over 70 to be vaccinated in high income countries (see, for example, Fig 2A, in which a substantial number of individuals aged 70–74 remain without booster vaccination in the UK under Strategy 1). In contrast, in our baseline analyses, when Strategy 1 is deployed in Sierra Leone, almost all older individuals receive booster vaccines (Fig 3A).

Different parameter sets led to variation in projected numbers of deaths. In S1A Fig, we show the range in the number of projected deaths in each country considered under Strategy 1. Crucially, despite the large variation in numbers of projected deaths between

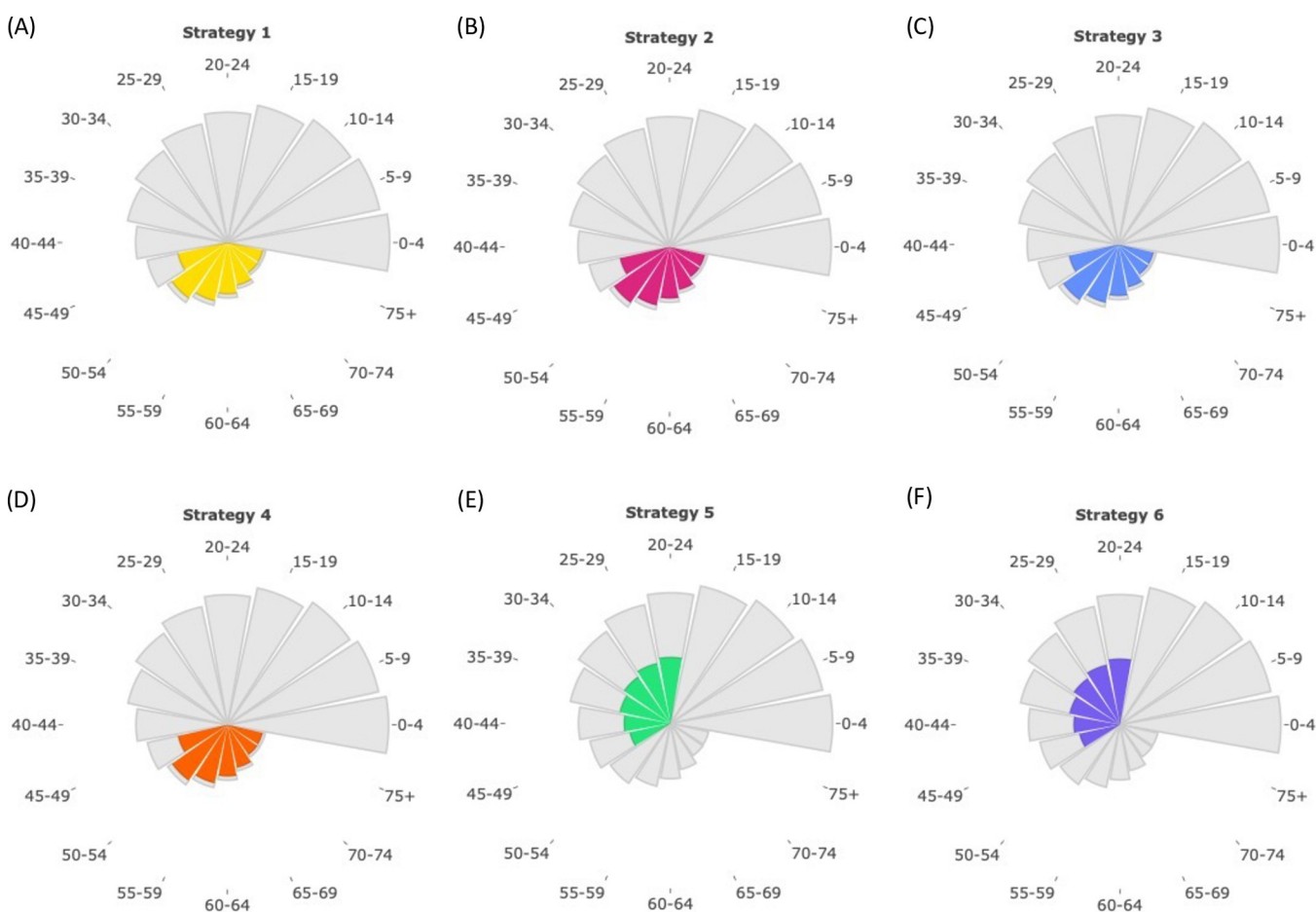

**Fig 3. The ages of individuals in Sierra Leone who receive booster vaccinations under each booster vaccination strategy.** The area of each wedge is proportional to the number of individuals in Sierra Leone in that age group. The coloured regions reflect the numbers of individuals in each age group who receive booster vaccinations under each booster vaccination strategy. The first four strategies generally involve prioritising older individuals for booster vaccination, whereas the final two strategies instead involve targeting younger individuals for booster vaccination. Booster vaccinations are administered at the start of our model projections.

parameter sets, for each parameter set individually, there were always fewer (or an equal number of) projected deaths when Strategy 1 was adopted compared to any of the other booster vaccination strategies (Fig 4B). This supports the use of a vaccination strategy involving directly protecting older individuals if a policy-maker aims to reduce the total number of deaths as much as possible.

In addition to projecting the number of deaths in each country under each booster vaccination strategy, we computed the projected YLL. This quantity not only accounts for the number of deaths, but also the ages of individuals when they die. As such, using booster vaccines to protect younger individuals may be expected to be of increased benefit if the goal is to minimise the expected YLL as opposed to overall mortality. However, in our model projections, we again found that Strategy 1 was the optimal strategy when YLL are considered (Fig 5), because of the substantial skew in the risk of severe outcomes of SARS-CoV-2 infection towards older individuals. We show the variation in the projected YLL between parameter sets under Strategy 1 in S1B Fig, but also note that for any individual parameter set, Strategy 1 always led to the fewest YLL across all strategies considered (Fig 5B).

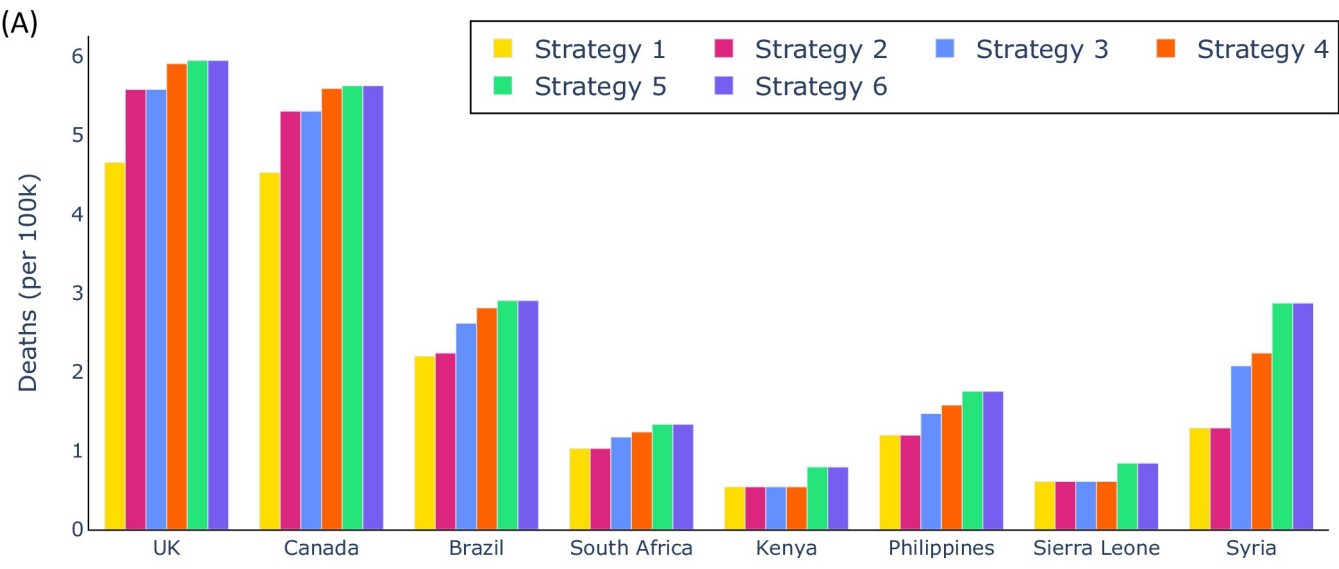

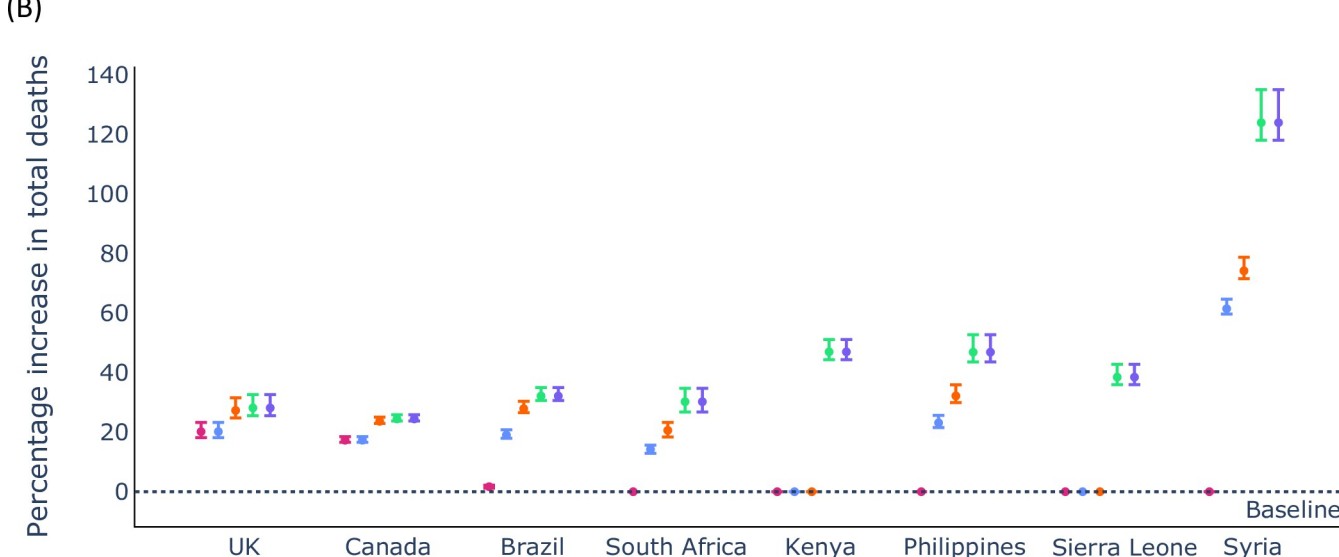

**Fig 4. Number of deaths (per 100,000 individuals) projected in different countries under different booster vaccination strategies.** A. The mean total number of deaths in each country under each vaccination strategy. B. The percentage increase in the total number of deaths under each vaccination strategy compared to under Strategy 1 (baseline). In panel B, the intervals represent 95% prediction intervals obtained across the 100 equally plausible parameter sets used in our analyses.

## Sensitivity analyses

In addition to our main analyses, we conducted further analyses in which we considered different assumptions regarding the number of available booster vaccinations (S2A and S2B Fig) and the maximum uptake of booster vaccination in each age group (S2C and S2D Fig).

We also generated model projections in which our baseline assumption that booster vaccines are administered at the start of our projections was relaxed. Specifically, in S3 Fig, we show the mean projected number of deaths in each country when the wave of the novel variant was assumed to have begun 150 days after booster vaccines were administered. Under those

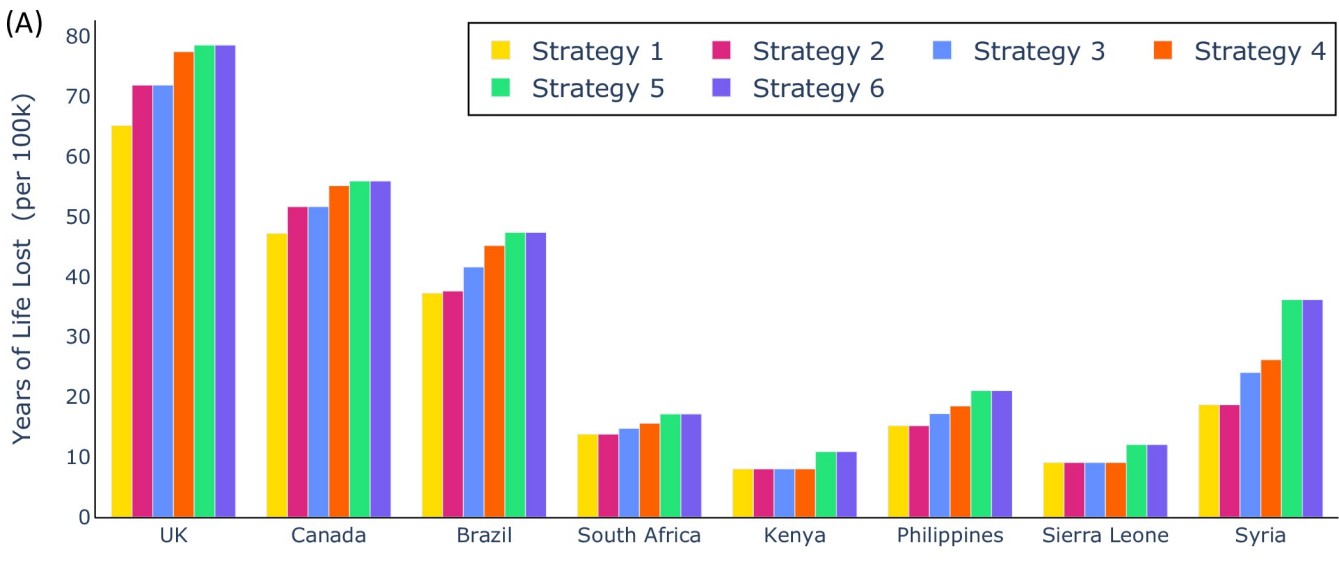

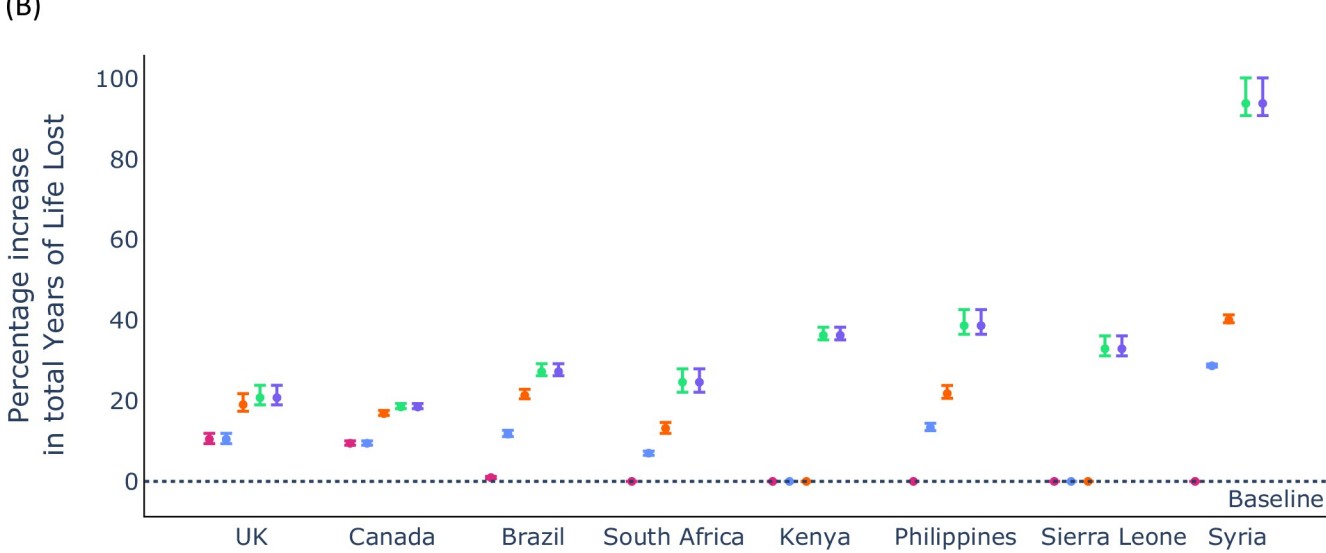

**Fig 5. Number of YLL (per 100,000 individuals) projected in different countries under different booster vaccination strategies.** A. The mean total YLL in each country under each vaccination strategy. B. The percentage increase in the total YLL under each vaccination strategy compared to under Strategy 1 (baseline). In panel B, the intervals represent 95% prediction intervals obtained across the 100 equally plausible parameter sets used in our analyses.

circumstances, fewer individuals were in the boosted class at the start of the outbreak as the immunity of many of the individuals who received a booster vaccine had waned. As a result, those individuals began the projection in the waned classes instead; we calculated the numbers of individuals in each class of the model at the start of our simulations by first running the model for 150 days in the absence of SARS-CoV-2 following the deployment of booster vaccines.

Crucially, in every case that we considered, we found that Strategy 1 led to fewer projected deaths and YLL than any of the other booster vaccination strategies.

## Discussion

Future SARS-CoV-2 transmission will be affected by a range of factors, including the emergence of novel variants, the extent to which booster vaccinations are administered and the dynamics of immunity. Compared to the early stages of the COVID-19 pandemic, many individuals globally have already been infected and vaccinated, with implications for population immunity and transmission. In this changing landscape, it is necessary to consider whether previous interventions are effective and whether different control strategies may now be optimal.

Here, we considered a wave of infections generated by a novel SARS-CoV-2 variant, and explored a range of different possible age-based booster vaccination prioritisation strategies. We considered both strategies that prioritise vaccinating older individuals (Strategies 1–4) and strategies that prioritise vaccinating younger individuals (who typically have more contacts; Strategies 5–6). We found that the implementation of an identical booster vaccination strategy in different countries can lead to very different compositions of the vaccinated population (*cf.* Figs 2 and 3). Direct protection of older individuals is easier to achieve with a limited number of booster vaccinations in countries with fewer older individuals, for example. This suggests that country-specific assessments of the effects of interventions may be required, particularly for precise quantitative projections to be made.

Despite differences in population structures between countries, in all of our analyses we found that age-based booster vaccination strategies that prioritise vaccinating the oldest individuals lead to more favourable public health outcomes than strategies that instead target younger individuals. This is because older individuals, if infected, are substantially more likely to experience severe outcomes of infection than younger individuals. This simple fact overwhelms any benefit that could be achieved by immunising individuals in age groups that are most responsible for transmission. As well as remaining true with different assumptions about booster vaccination supply, uptake and timing, we also found that this conclusion held for different objectives that are relevant to policy makers. Specifically, we considered scenarios in which the main objective is to reduce the total number of deaths due to SARS-CoV-2 in the host population, and those in which the objective is instead to reduce YLL.

As noted in the Introduction, this research builds on previous studies that seek to identify optimal vaccination strategies, particularly during the acute phase of the COVID-19 pandemic. While many previous studies have focussed on studying vaccination in the context of a single country, taken together they provide evidence about the impacts of vaccination in different locations. For example, Albani *et al.* [53] developed a model of SARS-CoV-2 transmission and vaccination in Chicago and New York City, and found that a vaccination campaign is expected to have a greater positive impact if it is initiated earlier in an outbreak. Similarly, Amaku *et al.* [54] modelled SARS-CoV-2 vaccination in Brazil and also found that delays to vaccination lead to higher mortality in that setting. Our research supports the conclusions of previous analyses that suggest that prioritising older and vulnerable individuals for vaccination is an essential part of a COVID-19 management strategy, particularly when there is limited vaccine supply. This conclusion has previously been reached in the context of a range of countries [55–58], although we note that other studies have hypothesised that prioritising individuals with large numbers of contacts for vaccination can sometimes be beneficial [26–29,59].

As with any epidemiological modelling analyses, the research presented here involved a range of assumptions and simplifications. For example, despite a huge amount of quantitative research having been undertaken, there remains uncertainty in the values of SARS-CoV-2 transmission parameters, the effectiveness of booster vaccines and the dynamics of waning immunity, particularly when considering novel SARS-CoV-2 variants. We assumed that

overall case fatality ratios differ between countries (these values were based on country-specific excess mortality data). However, the extent to which the risk of death varies between individuals of different ages was assumed to be identical in all countries, which may not reflect the impact and prevalence of comorbidities. Our baseline model projections also involved assumptions about the number of available booster vaccinations and vaccine uptake. We assumed that booster vaccines were administered at the start of our simulations in our main analyses, whereas in practice it would take time for booster vaccines to be administered. Importantly, however, we found our main qualitative conclusion–that prioritising the oldest individuals leads to the most favourable outcomes–was robust to consideration of other assumptions about booster vaccine availability, uptake and timing (S2 and S3 Figs).

A range of avenues exist for future research. While we used excess mortality data to estimate country-specific case fatality ratios, a more detailed analysis of the access to and quality of healthcare provision in different countries would be useful to assess the public health impacts of booster vaccination. It would also be interesting to consider different frequencies of booster vaccination, including assessing the impacts of annual or biannual booster vaccination campaigns. This may lead to more complex booster vaccination strategies becoming optimal (since, if an older individual has received a booster vaccination relatively recently, then it may be more effective to vaccinate a younger individual who has not yet received a booster vaccination). Future studies may also focus on the relative effectiveness of different types of booster vaccine–for example, comparing the use of a general SARS-CoV-2 booster vaccine against a variant-specific booster vaccine–or the allocation of booster vaccinations based on multiple risk factors rather than age alone [60].

In summary, in this study we have considered the impacts of different age-based booster vaccination strategies against SARS-CoV-2 in countries with different socio-economic backgrounds. This research updates previous work conducted in a scenario in which there was relatively little existing immunity in the host population. There remains substantial uncertainty about future SARS-CoV-2 transmission dynamics, particularly given the potential for novel variants to emerge with different characteristics. However, our research indicates that prioritising the oldest (and other vulnerable) individuals in host populations for booster vaccination is likely to be the optimal strategy in a wide range of different circumstances.

## Supporting information

**S1 Text. Supplementary details about the epidemiological model.** Information about the transmission model and its parameterisation.
(PDF)

**S1 Table. Parameter values used in our transmission model.** The final column indicates whether or not the value of each parameter varies between different parameter sets considered in our analyses.
(PDF)

**S1 Fig. Variation in the number of deaths and YLL between parameter sets under Strategy 1.** A. The number of deaths (per 100,000 individuals) projected in different countries under booster vaccination Strategy 1. Bars represent the mean values across all parameter sets (*cf*. yellow bars in Fig 4A in the main text), and lines indicate the 95% prediction intervals (reflecting variation between parameter sets). B. Analogous to panel A, but showing projected YLL rather than deaths. We note that, for any individual parameter set, Strategy 1 leads to the fewest projected deaths and YLL in every country (*cf*. Fig 4B and Fig 5B in the main text).
(TIF)

**S2 Fig. The impact of the number of available booster vaccines and the maximum uptake of booster vaccination on the mean number of deaths under each booster vaccination strategy.** Results are analogous to Fig 4A from the main text, but instead assuming: A. The number of available booster vaccines corresponds to 5% of each country's population size (rather than 10%). B. The number of available booster vaccines corresponds to 20% of each country's population size (rather than 10%). C. The maximum uptake of booster vaccination in any age group is assumed to be 50% (rather than 90%). D. The maximum uptake of booster vaccination in any age group is assumed to be 70% (rather than 90%). In each of these scenarios, Strategy 1 led to fewer (or an identical number of) projected deaths than any other booster vaccination strategy. We note that, in some circumstances, different booster vaccination strategies can lead to the same individuals being vaccinated (for example, as shown in Fig 3 of the main text, in our main analyses for Sierra Leone, Strategies 1–4 led to the same individuals receiving booster vaccinations). Here, strategies that led to the same individuals being vaccinated, and therefore identical projected numbers of deaths, are marked within each country and each panel using identical symbols above the relevant bars (for example, in panel A, in the UK strategies 2 and 3 led to the same individuals receiving booster vaccines and strategies 5 and 6 led to the same individuals receiving booster vaccines).
(TIF)

**S3 Fig. Number of deaths (per 100,000 individuals) projected in different countries under different booster vaccination strategies, for booster vaccines deployed 150 days before outbreak onset.** Results are analogous to Fig 4A in the main text, but with a delay of 150 days between booster vaccination and the beginning of the outbreak of the novel variant. The transmission model is first run in the absence of SARS-CoV-2 for 150 days to determine the impact of waning immunity prior to the arrival of the novel variant. The full transmission model (with the novel variant) is then run for a further 150 day period to determine numbers of deaths in different countries.
(TIF)

## Acknowledgments

Thanks to the World Health Organization for support that enabled us to undertake this research, including helpful feedback on our results. Thanks to members of the Zeeman Institute for Systems Biology and Infectious Disease Epidemiology Research at the University of Warwick, and the Wolfson Centre for Mathematical Biology at the University of Oxford, for useful discussions about this work.

## Author Contributions

**Conceptualization:** Edward M. Hill, Matt J. Keeling, Sam Moore, Robin N. Thompson.

**Formal analysis:** Ioana Bouros.

**Funding acquisition:** Robin N. Thompson.

**Investigation:** Ioana Bouros, Sam Moore.

**Methodology:** Edward M. Hill, Matt J. Keeling, Sam Moore, Robin N. Thompson.

**Project administration:** Robin N. Thompson.

**Supervision:** Robin N. Thompson.

**Validation:** Ioana Bouros.

**Visualization:** Ioana Bouros.

**Writing – original draft:** Ioana Bouros, Sam Moore, Robin N. Thompson.

**Writing – review & editing:** Ioana Bouros, Edward M. Hill, Matt J. Keeling, Sam Moore, Robin N. Thompson.

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
