## [Decision Letter · Decision Letter 0]

28 May 2024

Dear Dr Thompson,

Thank you very much for submitting your manuscript "Prioritising older individuals for COVID-19 booster vaccination leads to optimal public health outcomes in a range of socio-economic settings" for consideration at PLOS Computational Biology. As with all papers reviewed by the journal, your manuscript was reviewed by members of the editorial board and by several independent reviewers. The reviewers appreciated the attention to an important topic. Based on the reviews, we are likely to accept this manuscript for publication, providing that you modify the manuscript according to the review recommendations.

Sincerely,

Claudio José Struchiner, M.D., Sc.D.

Academic Editor

PLOS Computational Biology

Thomas Leitner

Section Editor

PLOS Computational Biology

Reviewer's Responses to Questions

**Comments to the Authors:**

Reviewer #1: The available vaccinations against SARS-CoV-2 protect against severe courses and death from the coronavirus, both on an individual and societal level. However, this protection is not permanent and therefore booster vaccinations are required.

The authors address the important question of which age groups should have access to booster vaccination and in what order to achieve an optimal effect. They consider different scenarios in several countries with the help of a mathematical model.

The methods used are appropriate, rigorous and detailed. The results are novel, relevant, and convincing. The discussion is comprehensive, fair, and balanced.

In approximately 20 years as a reviewer, this is the first time I have recommended accepting a manuscript in its original form. I would like to congratulate the authors on their excellent work.

Reviewer #2: It has been a pleasure to review the manuscript "Prioritising older individiuals for COVID-19-booster vaccination leads to optimal public health outcomes in a range of socio-economic settings".

The authors present a computational study of the outcomes (e.g. years of life lost) of different booster vaccination strategies (in terms of what population groups are prioritised), and apply the the approach to different socio-economic settings (from high to low income countries).

The underlying mathematical model is a deterministic, compartmental model of SEIR-type, with a chain of E compartments to model Gamma-distributed exposed times, and age-structure. It is a well-established and well-justified model of COVID-19.

Some of the parameter assumptions are quite strong. For example, assuming that the core infection parameter values are the same in all countries. However, the authors allowed for uncertainty in the parameters, and the results are robust to the uncertainty. However, it is not easy for the reader to discern what parameters are fixed and what parameters include uncertainty. Similarly, and although the robustness of the results is without question, the details on how some of the parameter models are fitted are not included in the manuscript.

Overall, this is an excellent study, and my comments below are just minor suggestions to improve the readibilty of the article.

Minor suggestions:

- Include confidence intervals in panel A of figures 4 and 5, even if they are small at that scale. Although the uncertainty is clear from panel B, it will help understand the robustness of the results.

- Describe clearly what parameters are taken as fixed, and what parameters include uncertainty.

- Describe the details of the MCMC estimates, even if it just in the supplementary material (for example, how many samples were generated, how was convergence assessed).

- In the supplementary material, Fig S1 suggests that in certain socio-economics settings (e.g. lower income countries), there are no significant differences between several strategies when the assumptions are relaxed (or alternatively, when more uncertainty is allowed in the assumptions). The overall conclusion of the paper in favour of Strategy 1 remains the same, but Fig S1 suggests than in certain settings the policy-makers can take a range of decisions about the strategy without affecting the final outcomes. Please consider including that observation in the main text.

**Have the authors made all data and (if applicable) computational code underlying the findings in their manuscript fully available?**

Reviewer #1: Yes

Reviewer #2: Yes

PLOS authors have the option to publish the peer review history of their article (what does this mean?). If published, this will include your full peer review and any attached files.

Reviewer #1: No

Reviewer #2: No

Figure Files:

Data Requirements:

Reproducibility:

References:

---

## [Editor Report · Decision Letter 1]

9 Jul 2024

Dear Dr Thompson,

We are pleased to inform you that your manuscript 'Prioritising older individuals for COVID-19 booster vaccination leads to optimal public health outcomes in a range of socio-economic settings' has been provisionally accepted for publication in PLOS Computational Biology.

Best regards,

Claudio José Struchiner, M.D., Sc.D.

Academic Editor

PLOS Computational Biology

Thomas Leitner

Section Editor

PLOS Computational Biology

---

## [Editor Report · Acceptance letter]

16 Jul 2024

PCOMPBIOL-D-24-00408R1 

Prioritising older individuals for COVID-19 booster vaccination leads to optimal public health outcomes in a range of socio-economic settings

Dear Dr Thompson,

I am pleased to inform you that your manuscript has been formally accepted for publication in PLOS Computational Biology. Your manuscript is now with our production department and you will be notified of the publication date in due course.

With kind regards,

Zsofia Freund
